# Internal Stress and Dislocation Interaction of Plate-Shaped Misfitting Precipitates in Aluminum Alloys

**DOI:** 10.3390/ma14195811

**Published:** 2021-10-04

**Authors:** Shinji Muraishi

**Affiliations:** Materials Science and Engineering, Tokyo Institute of Technology, Tokyo 152-8552, Japan; muraishi.s.aa@m.titech.ac.jp

**Keywords:** aluminum alloy, micromechanics, dislocation dynamics, Eshelby inclusion problem

## Abstract

The fine misfit precipitates in age-hardenable aluminum alloys have important roles due to their excellent age-hardening ability, by their interaction with dislocations. The present study focused on the internal stress field of plate-shaped misfitting precipitates to evaluate their roles in dislocation overcoming the precipitates by means of micromechanics based on Green’s function method. The stress field of misfit precipitates on {001} and {111} habit planes were reproduced by homogeneous misfit strain (eigenstrain) of the precipitate (Eshelby inclusion method), and the dislocation motion vector on the primary slip plane was predicted by the force acted on the dislocation by the Peach–Koehler formula. According to simulation results, the dislocation interaction strongly depends on the stress field and geometry of misfit precipitates; repulsive and attractive forces are operated on the dislocations lying on the primary slip plane when the dislocation approaches the misfit precipitates. The hardening ability of different orientations of precipitation variants was discussed in terms of interaction force acted on the dislocation.

## 1. Introduction

The fine precipitation microstructure in age-hardenable aluminum alloys has an important role due to its excellent age-hardening ability by dislocation-precipitate interactions; therefore, the microstructural design of precipitates in aluminum alloys has attracted much attention over many decades. Precipitation microstructure at the peak aging condition consists of the Guinier–Preston zone (GP-zones) and meta-stable phases on certain habit planes in typical age-hardenable aluminum alloys, e.g., {001} GP zones and θ” plates in Al-Cu, {001} β” needles in Al-Mg-Si, spherical GP-zones and {111} η” in Al-Zn-Mg alloys, etc. A further strengthening effect is achieved by the modification of the precipitation sequence, morphology and interfacial structure of precipitates with the small addition of trace elements into alloy systems [1,2]. A recent review paper for precipitation in nanostructured alloys pointed out that the fundamentals behind the coupling of dislocations and precipitates need to be further investigated [3].

The strengthening mechanisms of fine precipitates are classified into (I) misfit hardening and (II) Orowan by-passing mechanisms, where the aging condition of peak hardness is chosen depending on the size and magnitude of the misfit strain of precipitates under a certain volume fraction of precipitates. In general, misfit hardening is the major hardening mechanism by which fine shearable precipitates are formed at the early and medium stages of aging conditions; on the contrary, the Orowan mechanism is operated at the latter stage of the aging condition for relatively large precipitates, which are hard to cut by dislocation. Regarding misfit strain hardening, since the internal stress field of a precipitate varies depending on the misfit strain components and the shape of the precipitate, the strengthening ability of the precipitate would be varied with the interaction manner of the dislocation and the precipitate. Our previous work revealed that strengthening by {001} GP zone in an Al-Cu single crystal shows a strong temperature and orientation dependence in yield and flow stress, whereas the strengthening effect of {001} GP-zone is emphasized by the dislocation of the Burgers vector parallel to the GP-zone plate at a low temperature [4,5]. A recent molecular dynamics simulation predicted the temperature and strain rate dependency on the cutting and Orowan-looping mechanisms in a dislocation-GP zone interaction [6]. In view of the interaction energy analysis between the misfit precipitate and dislocation, the result seems contradicted by the general concept of misfit hardening because the elastic interaction is absent in the case of the dislocation overcoming (001) GP-zone with the Burgers vector parallel to the GP-zone plate. According to atomistic simulation results reported by Singh et al. [7,8], the orientation dependency of strengthening by the GP-zone strongly depends on the screw and edge character of dislocation; hereby, the importance of cross-slip events on the precipitation hardening is pointed out.

A recent development in atomistic and mesoscopic simulations provides the calculation method of the local dislocation interaction with the precipitate, and much effort has been spent on the prediction of macroscopic stress-strain responses. For this purpose, a computational framework based on micromechanics is beneficial to determine the internal stress field caused by defects in the considered material, e.g., point defects, dislocations, second particles, etc. The advantage of this method is that the stress field of the dislocation and precipitate is directly computed by a simple integral expression of Green’s function without solving the boundary value problem, thus the superposition principle holds in the spatial and average stress fields given by Eshelby’s inclusion method and the Mori–Tanaka mean field theory, etc. [9,10,11]. Among internal stress fields, the dislocation motion is readily predicted by computing the force acted on the dislocation (Peach–Koehler force); the dislocation reactions and interactions can be reproduced by a computational algorithm. The present study focused on the internal stress field of plate-shaped misfitting precipitates in aluminum alloys, especially the strengthening effect of {001} and {111} plate-shaped misfit precipitates were considered in terms of the elastic interaction between the dislocation and the misfit precipitate. The stress field of misfit precipitates of aluminum alloys on a certain habit plane was reconstructed with the homogeneous misfit strain (eigenstrain) inside the precipitate, and then the dislocation motion was simulated in accordance with the force acted on the dislocation. The orientation dependency of the hardening ability of {001} and {111} plates was discussed in terms of the geometrical difference in dislocation cutting manners.

## 2. Theory of Micromechanics and Eigenstrain Formulation

The dislocation motion under an external stress field is greatly influenced by the internal stress field of the precipitate and dislocations. The spatial distribution of stress caused by internal stress origins is readily computed by Green’s function method based on micromechanics, which is well documented in textbook [9]. First, total stress, $\sigma_{ij}^{T}$, is represented by the sum of external stress $\sigma_{ij}^{0}$ and internal stress $\sigma_{ij}^{Int}$ as follows:(1)$$\sigma_{ij}^{T}=\sigma_{ij}^{0}+\sigma_{ij}^{Int}=\sigma_{ij}^{0}+\sigma_{ij}^{D}+\sigma_{ij}^{\Omega }.$$
where $\sigma_{ij}^{0}$ is external stress without the disturbance of internal stress, $\sigma_{ij}^{D}$ is internal stress caused by dislocations, $\sigma_{ij}^{\Omega }$, is internal stress caused by precipitates. Note that the internal stress field in Equation (1) varies depending on the geometry of the dislocation segments and precipitates. Since dislocation motion is predicted by the stress acted on the dislocation, the stress computation is performed only on the dislocation nodal points. This indicates that the elastic field of the entire body of material is not needed in the present simulation, so that dislocation motion can be predicted analytically without solving the boundary value problem, such as with FE analysis.

### 2.1. Stress Field of Precipitate

According to the Eshelby inclusion theory [9,10,11,12], the stress of the precipitate $\sigma_{ij}^{\Omega }$ at the field point x can be given through the Eshelby tensor as follows:(2)$$\sigma_{ij}^{\Omega }\left(x\right)=C_{ijkl}\left(S_{klmn}\varepsilon_{mn}^{*}-\varepsilon_{kl}^{*}\right)\;.$$
where $S_{ijkl}$ is Eshelby tensor, $C_{ijkl}$ is stiffness tensor, $\varepsilon_{mn}^{*}$ is eigenstrain of region Ω. Supposing that the region Ω is subjected to uniform eigenstrain, the elastic field inside Ω becomes uniform when an ellipsoidal inclusion is assumed. The analytical solution of the Eshelby tensor for isotropic ellipsoidal inclusion is listed in textbook [9], while the arbitrary shape of the inclusion should be computed numerically by Green’s function. Furthermore, the analytical solution for the stress outside Ω is limited for the isotropic case [13], so that the exterior point stress should be computed also numerically. Assuming that the misfit strain is the major internal stress origin (inclusion problem), the superposition principle of stress holds even in the case of multiple inclusions. The general integral expression of inclusion stress is represented by the surface integral as follows:(3)$$S_{klmn}^{}\left(x\right)=-\int_{\left|\Omega \right|}^{}C_{pqmn}\left(\frac{G_{kp,l}\left(x,x^{\prime }\right)+G_{lp,k}\left(x,x^{\prime }\right)}{2}\right)n_{q}dS\left(x^{\prime }\right)$$
where *G_ij,k_* is the first derivative of Green’s function, n_k_ is outward vector normal to |Ω| for the interior point Eshelby tensor, while the inward direction is for exterior point Eshelby tensor. Note that Equation (3) gives the total strain at the region inside and outside, then eigenstrain should be subtracted as shown in Equation (2) when the evaluation point is taken at the interior point of Ω. In the case that the stiffness of inclusion is different from that of the matrix phase, the stress disturbance should be considered as an inhomogeneity problem. In such a case, the fictitious eigenstrain assumed in Ω is proportionally changed with the stress caused by other stress origins, such as dislocation, inhomogeneity, external stress, etc. Nevertheless, the eigenstrain formulation of the present study is still valid whenever the dislocation interacts with the elastic field of the misfit precipitate.

### 2.2. Dislocation Stress

The stress field of dislocation is computed by line integral with respect to dislocation segment as follows [7]:(4)$$\sigma_{ij}^{D}\left(x\right)=C_{ijkl}\int_{L}^{}\epsilon_{lnh}C_{pqmn}G_{kp,q}\left(x,x^{\prime }\right)b_{m}\nu_{h}dl\left(x^{\prime }\right)\;.$$
where $C_{ijkl}$ is stiffness tensor, $G_{ij,k}$ is first derivative of Green’s function, $b_{i}$ is Burgers vector, ϵ_*ijk*_ is permutation tensor, $\nu_{i}$ is tangent vector of dislocation line. Specifically, the numerical Gauss integration technique is taken along a discretized dislocation line segment by computational method, where parametric shape functions are used for representation and interpolation of the position vector and field quantities [10,14].

### 2.3. Peach-Koehler Force

Once the internal stress fields of dislocations and precipitates are obtained, the force acted on the dislocation can be computed by the Peach–Koehler formula as follows [9,10,15]:(5)$$f_{m}=\epsilon_{jmn}\sigma_{ij}b_{i}\nu_{n}\;.$$
where $f_{i}$ is force vector and $\nu_{i}$ is tangent vector of the dislocation line. It is noted that the force vector is always normal to the dislocation line, while the direction of dislocation motion should have taken place on a certain slip plane. In the present study, the stress calculation is performed on the orthogonal crystalline coordinates of a face-centered cube (fcc), then the resultant Peach–Koehler force (PK force) is resolved into three orthogonal vectors associated with the local coordinate of {111} dislocation slip system. A velocity vector proportional to the force vector is set to be parallel to the slip direction, then displacement of dislocation nodal points are updated after a certain time period. For estimating the hardening ability of a precipitation variant, interaction forces acted on the glide dislocations with different cutting manners are mutually compared.

### 2.4. Energy Consideration of Precipitation Hardening

The evolution of dislocation microstructure under external stress is influenced by the dislocation motion among local internal stress fields of dislocations and precipitates, where the total plastic strain by dislocation motion should be treated as the macroscopic average of strain. First, the amount of plastic work performed on the material is mentioned by
(6)$$\delta W=\int_{V}^{}\sigma_{ij}^{T}\delta \varepsilon_{ij}^{\;P}dV=V\sigma_{ij}^{T}\langle \delta \varepsilon_{ij}^{\;P}\rangle \;\;\;$$
where δW is plastic work performed on the material, $\delta \varepsilon_{ij}^{P}$ is variation in plastic strain, $\sigma_{ij}^{T}$ is total stress, respectively. However, due to the fact that the internal stress is zero in average, $\sigma_{ij}^{T}$ in the entire volume of material equals external stress on the loading surface, $\sigma_{ij}^{0}$.

Accounting that the increment of plastic strain is mentioned by the total area swept by dislocations in considered volume, the discrete dislocation slip can be translated into the averaged plastic strain. According to Mura [9], the amount of plastic strain in considered volume, V, can be mentioned by the product of Burgers vector, $b_{i}$, and the area swept by dislocations, $n_{j}\delta$*S,* as follows:(7)$$\langle \delta \varepsilon_{ij}^{\;P}\rangle =\frac{1}{V}\int_{\delta S}^{}b_{i}n_{j}\delta S=\frac{1}{V}\int_{L}^{}b_{i}\epsilon_{jmn}\delta u_{m}\nu_{n}dl\;\;\;\;\;$$
where small displacement of dislocation motion $\delta u_{m}$ is assumed in line integral. By substituting Equation (7) into Equation (6), plastic work performed on the material is mentioned by that mediated by total dislocation motions in material as follows:(8)$$\delta W=\int_{L}^{}\sigma_{ij}^{T}b_{i}\epsilon_{jmn}\delta u_{m}\nu_{n}dl=\int_{L}^{}f_{m}^{T}\delta u_{m}dl\;\;$$

In order to discuss the elastic energy and its interactions associated with the dislocation and precipitate, Gibbs free energy change is considered. the total Gibbs free energy associated with external stress, precipitates and dislocations is prescribed by elastic strain energy, E, and potential energy, P, as follows:(9)$$G=E^{S}+E^{I}-P\;\;\;.$$
where $E^{S}$ is elastic energy of external stress, dislocations, and precipitates without interactions between them (self-energy), which is reduced to the following:(10)$$E^{S}=\frac{1}{2}\int_{V}^{}\sigma_{ij}^{0}e_{ij}^{0}dV+\frac{1}{2}\underset{p,m}{\sum }\int_{V}^{}\left(\sigma_{ij}^{\Omega \left(p\right)}e_{ij}^{\Omega \left(p\right)}+\sigma_{ij}^{D\left(m\right)}e_{ij}^{D\left(m\right)}\right)dV.$$
where superscript $\Omega \left(p\right)$ and $D\left(m\right)$ indicates stress associated with *p*-th inclusion and m-th dislocation, respectively. Summation is taken with respect to p and m. Meanwhile $E^{I}$ is interaction energy between precipitates and dislocations, as follows:(11)$$E^{I}=\underset{p,q\left(p\neq q\right)}{\sum }\int_{V}^{}\sigma_{ij}^{\Omega \left(p\right)}e_{ij}^{\Omega \left(q\right)}dV+\underset{m,n\left(m\neq n\right)}{\sum }\int_{V}^{}\sigma_{ij}^{D\left(m\right)}e_{ij}^{D\left(n\right)}dV\;+\underset{p,n}{\sum }\int_{V}^{}\sigma_{ij}^{\Omega \left(p\right)}e_{ij}^{D\left(n\right)}dV$$
where the first term is the interaction of precipitates and the second term is interaction of dislocations. The third terms are dislocation–precipitate interactions, which are expressed by dislocation stress and precipitate strain. Potential energy is defined by tractions and displacement on the external loading surface, $S_{0}$, which is as shown below: (12)$$P=\sum_{q,n}\int_{S_{0}}^{}\sigma_{ij}^{0}\left(u_{i}^{0}+u_{i}^{\Omega \left(q\right)}+u_{i}^{D\left(n\right)}\right)n_{j}dS.$$

The first term is potential energy due to external loading and the second and third terms are the interaction of external stress and eigenstrain associated with precipitates and dislocations. Note that interaction between external stress and internal stress is absent in the elastic energy of Equation (11) (Collonetti’s theorem [9]), while external stress interacts with internal stress in potential energy in Equation (12). Superscript $\Omega \left(n\right)$ and $D\left(n\right)$ indicate n-th inclusion and dislocation segment, respectively. Since the elastic strain of the internal stress origin should be zero at the traction-free surface, $u^{\Omega }$ and $u^{D}$ are regarded as the displacement caused by average eigenstrain in the whole volume. Note that although the energy associated with the precipitate and dislocation in Equation (10)–(12) is computed by the volume integral of the entire volume, the integral region is eventually reduced to the region of eigenstrain due to the property of internal stress. By translating volume integrals associated with dislocation motion in Equation (10)–(12) into surface integrals, the energy associated with the dislocation motion can be obtained as follows:(13)$$E^{S}=\frac{1}{2}\int_{V}^{}\sigma_{ij}^{0}e_{ij}^{0}dV-\frac{1}{2}\underset{p}{\sum }\int_{\Omega \left(p\right)}^{}\sigma_{ij}^{\Omega \left(p\right)}\varepsilon_{ij}^{*\Omega \left(p\right)}dV-\frac{1}{2}\underset{m}{\sum }\int_{S\left(m\right)}^{}\sigma_{ij}^{D\left(m\right)}b_{i}^{D\left(m\right)}n_{j}dS$$
(14)$$E^{I}=-\underset{p,q\left(p\neq q\right)}{\sum }\int_{\Omega \left(q\right)}^{}\sigma_{ij}^{\Omega \left(p\right)}\varepsilon_{ij}^{*\Omega \left(q\right)}dV-\underset{p,n}{\sum }\int_{S\left(n\right)}^{}\sigma_{ij}^{\Omega \left(p\right)}b_{i}^{*D\left(n\right)}n_{j}dS\;-\underset{m,n\left(m\neq n\right)}{\sum }\int_{S\left(n\right)}^{}\sigma_{ij}^{D\left(m\right)}b_{i}^{D\left(n\right)}n_{j}dS$$
(15)$$P=\sigma_{ij}^{0}\left(\int_{V}^{}e_{ij}^{0}dV+\sum_{q}\int_{\Omega \left(q\right)}^{}\varepsilon_{ij}^{*\Omega \left(q\right)}dV+\sum_{n}\int_{S\left(n\right)}^{}b_{i}^{D\left(n\right)}n_{j}dS\right).$$

For simplicity, the energy difference due to the dislocation motion is considered by assuming the small displacement of dislocation segments in Equation (13)–(15). Variation in Gibbs free energy associated with dislocation motion is mentioned by $\delta G=G\left(dS+\delta S\right)-G\left(dS\right)$ as follows:(16)$$\delta G=\delta E^{S}+\delta E^{I}-\delta P.$$

Therefore, by using the translation property of surface and line elements, $n_{j}\delta S=\epsilon_{jmn}\delta u_{m}\nu_{n}dl$, variation form of energy components become the following;
(17)$$\delta E^{S}=-\frac{1}{2}\sum_{t}\int_{L\left(t\right)}^{}\sigma_{ij}^{D\left(t\right)}b_{i}^{\left(t\right)}\epsilon_{jmn}\delta u_{m}^{\left(t\right)}\nu_{n}dl\approx \;\delta E^{core}.$$
(18)$$\delta E^{I}=-\sum_{t}\int_{L\left(t\right)}^{}\left(\sum_{p}\sigma_{ij}^{\Omega \left(p\right)}+\sum_{s\left(s\neq t\right)}\sigma_{ij}^{D\left(s\right)}\right)b_{i}^{\left(t\right)}\epsilon_{jmn}\delta u_{m}^{\left(t\right)}\nu_{n}dl.$$
(19)$$\delta P=\sigma_{ij}^{0}\left(\sum_{t}\int_{L\left(t\right)}^{}b_{i}^{\left(t\right)}\epsilon_{jmn}\delta u_{m}^{\left(t\right)}\nu_{n}dl\right).$$
where $\delta E^{S}$, $\delta E^{I}$, δP is variation in dislocation self-energy, interaction-energy and potential energy, respectively. Note that, due to the singularity of the self-stress in the dislocation core, the self-energy of dislocation is often alternated by line tension approximation $\delta E^{core}$. By taking the sum of energy differences of external and internal origins, variation in Gibbs free energy becomes the following;
(20)$$\delta G=-\underset{t}{\sum }\int_{L\left(t\right)}^{}\left(\sigma_{ij}^{0}+\underset{p}{\sum }\sigma_{ij}^{\Omega \left(p\right)}+\underset{s}{\sum }\sigma_{ij}^{D\left(s\right)}\right)b_{i}^{\left(t\right)}\epsilon_{jmn}\delta u_{m}^{\left(t\right)}\nu_{n}dl$$

Eventually, the stress mentioned in Equation (20) is the total stress acted on dislocation segments. Note that the numerical sum of $\sigma_{ij}^{D\left(s\right)}$ accounts for all the dislocation segments, where interaction stress (s ≠ t) is computed by Equation (4), while self-stress (s = t) can be alternated by line tension approximation. By comparison of Equation (20) to the macroscopic external plastic work in Equations (6) and (8), the second and third terms in Equation (20) are regarded as the change in interaction energy. Although the average internal stress should be zero in the entire volume of material ($\sigma_{ij}^{T}=\sigma_{ij}^{0}$), the dislocation motion is influenced by the internal stress field of dislocations and precipitates because the total sum of interaction stress at the dislocation nodal point is locally not zero. Recalling the fact that the dislocation glide motion and total force vectors should be the same directions, δG is always decreased whenever dislocation motion takes place such as in the following: (21)$$f_{m}^{T}\delta u_{m}=\left(f_{m}^{0}+f_{m}^{\Omega }+f_{m}^{D}\right)\delta u_{m}>0.$$

Accordingly, depending on the signs and the magnitude of interaction forces, external and internal work is classified into the following three cases: (I) hardening behavior, (II) softening behavior, (III) annihilation behavior. In case (I), the direction of interaction force and external force is opposite $\left(f_{m}^{T}<f_{m}^{0}\right)$, then the dislocation motion is retarded and the amount of δG becomes smaller since the amount of plastic displacement is decreased. In case (II), in contrast to the above, the total force acted on the dislocation is increased by internal stress $\left(f_{m}^{T}>f_{m}^{0}\right)$; therefore, the dislocation motion is promoted and the amount of δG is negatively increased with the increase in plastic displacement. In case (III), annihilation of dislocations takes place due to the negative direction of strong interaction forces since the displacement vector turns into negative direction. This situation may be made possible by the dislocation motion driven by the strong internal stress of the precipitate. In all the cases, δG is negatively increased by the dislocation motion, but the amount of $\delta u_{m}$ is greatly influenced by the internal stress field, hence the interaction force acted on the dislocation is important to be clarified in a later section.

### 2.5. Simulation Method

An analysis of the dislocation interactions with precipitation variants in aluminum alloys was made for {001} plates and {111} plates. Linear elasticity was assumed for the aluminum matrix and precipitate phases, where the anisotropic Green’s functions of aluminum were used for the computation of dislocation and precipitate stresses [11]. The stiffnesses of aluminum are, $c_{11}=108.2,\;c_{12}=61.3,\;c_{44}=28.5$ GPa. The dislocation interaction with different orientations of {001} and {111} precipitation variants were demonstrated by (111) [−110] dislocation slip system, where two and three types of dislocation cutting manners exist among three {001} and four {111} variants, respectively. The initial dislocation arrangement was started from the straight screw dislocation, whose ending points with the length of 1732 nm were fixed to simulate the Frank–Read source. In order to clarify the geometrical effect of the internal stress of {001} and {111} variants on the interaction force, the computational model of a (001) plate was prepared by an ellipsoidal inclusion with 200 nm in radius and 50 nm in thickness. By the coordinate transformation of the spatial position and eigenstrain components of the model, the stress fields of {001} and {111} variants with different orientations were demonstrated in the global coordinate of fcc crystal by Equations (2) and (3). Based on the lattice constant of θ” plate in Al-Cu alloy [16], the non-zero component of eigenstrain normal to the plate was set to be $\varepsilon_{33}^{*}$ = 0.04, which was assumed in the local coordinate of the plate. The dislocation motion calculation was made through Equations (4) and (5), the velocity of dislocation was proportionally changed with the total force acted on the dislocation segments. In the present study, the glide motion of dislocation on (111) slip plane was only considered by using the resolved force vector along the direction parallel to the slip plane under the condition of constant applied stress of 4.0 GPa along [001] crystal orientation (resolved shear stress, 1.63 GPa). The simulation results considering the cross-slip event can be found in our recent report [17].

## 3. Dislocation Dynamics Simulation

### 3.1. Internal Stress Field of {001} and {111} Plate

According to Eshelby’s inclusion theory, the stress field inside and outside the inclusion is shape-dependent, which can be semi-analytically computed by Green’s function method. Therefore, the effect of a different orientation of the precipitation variant on the dislocation motion is evaluated by employing the same computation model of the precipitate. Since the principle axis of inclusion in the global coordinate is varied with the orientation of the variant, the unidirectional eigenstrain component in the local coordinate of inclusion is transformed into the global coordinate for the stress calculation of the precipitate. Figure 1 shows the stress field obtained by the present method for {111} plate in the local coordinate. The horizontal axis indicates a normalized distance from the origin of inclusion, where the interface of the precipitate and matrix is at $D/R_{c}=1.0$. Since uniaxial eigenstrain $\varepsilon_{33}^{*}$ = 0.04 is assumed in the local coordinate, stress $\sigma_{33}^{}$ is the major stress component with respect to the axis normal to the plate. Since anisotropic Green’s function of aluminum was employed in the present study, the stress value inside the inclusion was compared with the analytical solution of Eshelby’s tensor for ellipsoidal inclusion with a similar shape (aspect ratio, $R_{c}/R_{a}$ = 0.25). The analytical solution of the stress inside the inclusion yields $\sigma_{33}=-0.608,\;\sigma_{11}=\sigma_{22}=-0.382$ GPa, where Eshelby’s tensor of oblate spheroid is calculated to be $S_{1133}=S_{2233}=-9.27\times 10^{-3},\;S_{3333}=0.870$ (Poisson’s ratio, υ=1/3). Therefore, the stress values in Figure 1 are valid and consistent with analytical solutions.

### 3.2. Dislocation Interactions with {111} Variants

The dislocation interaction with a plate-shaped precipitate on {001} and {111} habit planes is computed by the dislocation dynamics simulation. Since the interaction manners in {001} plates are basically similar to {111} plates, except when the intersecting manner of (111) variant is parallel to the slip plane, the results obtained for {111} plates are represented. There are three types of dislocation interactions in {111} plates, which are classified as Type-A, Type-B, and Type-C. Figure 2 shows the following snapshots of dislocation interactions with {111} precipitates: (a) Type-C: (111) plate parallel to slip plane, (b) Type-B: (−111) plate cut by 60 deg., (c) Type-B: (1−11) plate cut by 60 deg., and (d) Type-A: (11−1) plate cut by 0 deg. Dislocation of the slip system and the tangent vector of initially straight dislocation is common, and the different orientation of {111} plates are placed at the same position in all the cases. The magnitude and direction of the interaction forces caused by {111} plate, which include glide and climb forces, are represented by arrows in the same magnitude of scale. It is apparent that the interaction force acted on the dislocation is minutely small in (a) and (d), while strong interaction forces are operated in (b) and (c). Since the directions of force vector in (b) and (c) are opposite under the same dislocation configuration (e.g., dislocation motion vector, slip plane, tangent vector, Burgers vector), the existence of a strong orientation dependency in interaction manners of dislocation and {111} plate is suggested by the present simulation.

### 3.3. Interaction Force Analysis of {111} Variants

Line profiles of the interaction force acted on a single nodal point of screw dislocation during the dislocation motion are plotted in Figure 3. Since the magnitude of interaction forces become quite small values in the case of dislocation interaction with (111) and (11−1) plates as shown in Figure 2, the changes in the force profiles for (−111) and (1−11) plates are plotted in Figure 3a,b, respectively. The values of the interaction forces normalized by the magnitude of Burgers vector indicate the shear component of internal stress acted on the dislocation, where the positive and negative signs indicate the force promoting and retarding the dislocation motion. In the case of dislocation interaction with (−111) plate in (a), the dislocation motion is retarded at first by the negative interaction force operated on the dislocation in front of the (−111) plate (at $D/R_{c}=-5$), which is consistent with the force vector as represented in Figure 2b. By further dislocation motion, the negative interaction force is turned into a positive one to promote the dislocation motion toward the advancing side. It can be seen that the interaction force profile is symmetric with respect to the origin of (−111) plate (at $D/R_{c}=0$). By comparing the force profile of (−111) plate in (a) and (1−11) plate in (b), the signs of the interaction forces are opposite, but the profiles are almost similar in shape and magnitude. As shown in Figure 3b, promoting and retarding forces are operated outside and inside the (1−11) plate, respectively (the region inside the plate is within $D/R_{c}<\pm 3$). Accounting for the dislocation interactions of {111} variants as shown in Figure 3, averaged interaction energy should be reduced to zero since the misfit plates with positive and negative interactions are equally intersected by the same character of dislocation. However, since the stress necessary for the dislocation overcoming {111} variants is orientation-dependent, it is natural to consider that the interspacing for the dislocation bowing-out stress is changed with the geometry of the dislocation and precipitation variants. Note that when the tangent vector of dislocation is inversed, the signs of PK forces associated with the external and internal stresses are inversed. Therefore, the hardening abilities of Type-B variants are irrespective of the sign of screw dislocation. On the other hand, due to a similar reason, the relation of the strong and weak strengthening by Type-B variants can be alternated when the sign of the external stress is inversed. However, recalling the strengthening by the stress-oriented GP-zones of Al-Cu single crystalline by Eto [4], the strengthening by Type-B variants always exhibited a smaller stress level under the tensile and compressive applied stress as compared with the Type-A variant. This situation can be made possible by positive and negative stresses in front of and behind the precipitate since the weak strengthening orientation always exists when the dislocation tangent vector is inversed. Within the context of the present study, the weak strengthening effect by the Type-B variant can be partially explained by taking into account the PK force acted on the screw dislocation on the primary slip plane without the cross-slip event. Nevertheless, it is evident that there exist strong and weak strengthening effects by Type-B variants, hence a favourable orientation relationship exists in dislocation overcoming the {001} and {111} plates among the internal stresses of misfitting precipitates.

## 4. Conclusions

The micromechanical analysis of the stress field caused by {001} and {111} misfit precipitates in an aluminum alloy, and their interactions with dislocations, was performed by Green’s function method. The numerical computation results of the stress field of {001} and {111} plates obtained by the present study show good consistency with the analytical solution of Eshelby’s method. As a result of the computation of the interaction force acted on the dislocation, the dislocation interaction with the misfit plate is absent when the dislocation Burgers vector intersects parallel to the plate for {001} and {111} misfit precipitates (Type-A and Type-C variants). Meanwhile, when the Burgers vector intersects the plate by 60 deg. (Type-B variant), positive and negative interactions appeared, depending on the orientation of variants, and signs of interaction forces were alternated by the dislocation motion directions under external stress. Therefore, the strengthening by the misfit precipitate is orientation dependent, its hardening ability can be changed with the motion direction of dislocation against the precipitate. In view of external work performed on the material, dislocation motion behaviour is greatly influenced by the existence of internal stress of the misfit precipitate.

## Figures and Tables

**Figure 1 materials-14-05811-f001:**
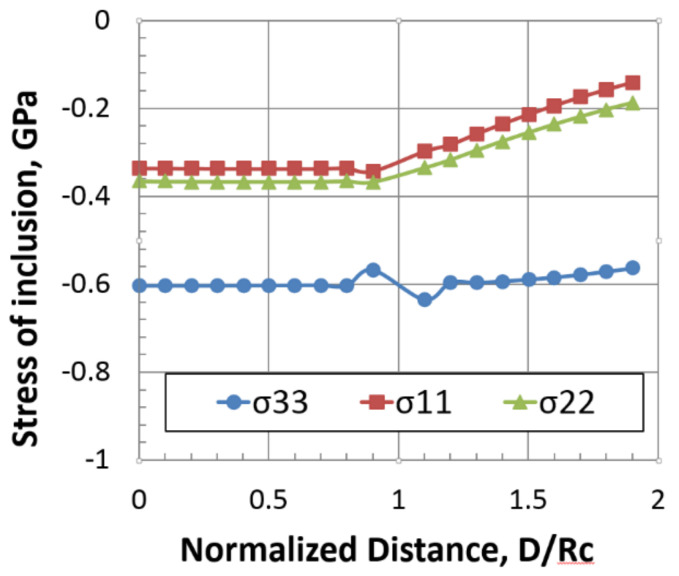
Line profile of inclusion stress along plate normal direction. Uniaxial positive misfit strain normal to the plate is assumed. Center of the inclusion is indicated by D/*R_c_* = 0.0, interface of precipitate and matrix by D/*R_c_*= 1.0.

**Figure 2 materials-14-05811-f002:**
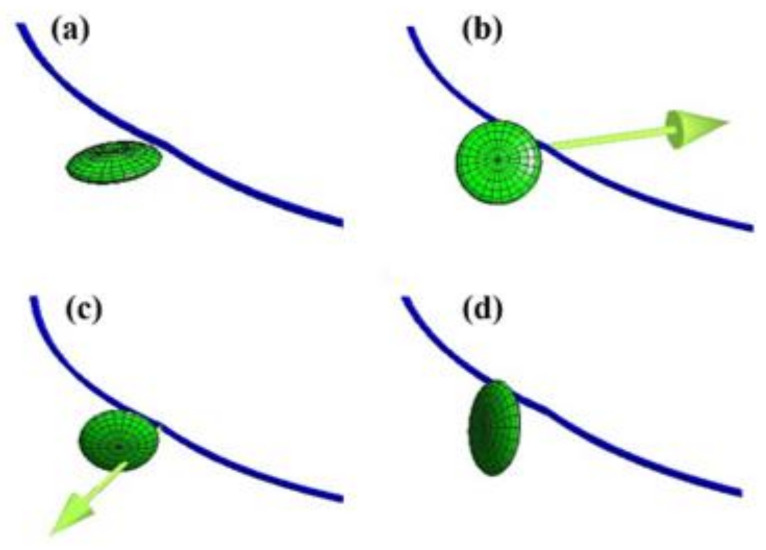
Dislocation interactions with different orientation of {111} plates. (**a**) Type-C: (111) plate parallel to slip plane, (**b**) Type-B: (−111) plate intersected by 60 deg., (**c**) Type-B: (1−11) plate intersected by 60 deg., (**d**) Type-A: (11−1) plate intersected by 0 deg. Arrow indicating interaction force acted on the dislocation nodal point. Direction of dislocation motion under external stress is from upper right to lower left. Interaction forces are absent in (**a**,**d**), while strong interaction forces with opposite directions in (**b**,**c**).

**Figure 3 materials-14-05811-f003:**
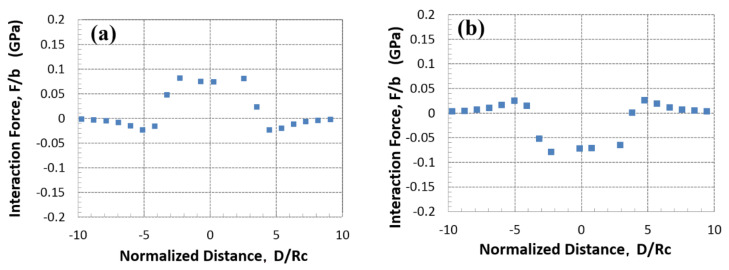
Interaction force acted on screw dislocation intersecting (−111) plate in (**a**), (1−11) plate in (**b**). Distance from the origin of {111} plate is normalized by radius Rc.

## Data Availability

Not applicable.

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
