# Peer review of "Internal Stress and Dislocation Interaction of Plate-Shaped Misfitting Precipitates in Aluminum Alloys"

_materials, 2021, doi:10.3390/ma14195811_

Round 1

Reviewer 1 Report

The authors present an analysis of the elastic field and energy of an Eshelby inclusion near a dislocation. They then provide a few numerical results intended to simulate precipitation strengthening in aluminum alloys. Overall I find the contribution here far too meager to constitute a journal article. The theoretical analysis in Section 2 is textbook material (see Mura for example), there is no new contribution there. The numerical results are very thin, no significant insights can be gained. Many details on parameter values, numerical methods, analytical solutions, etc. are missing. This article needs to be expanded significantly (perhaps by a factor of 10 to 100) and the quality greatly improved before it is suitable for publication anywhere.

Reviewer 2 Report

This work performs dislocation theory-based calculations to show that glide dislocation – precipitate platelet interaction depends on precipitate orientation. When Burgers vector intersects parallel to the precipitate plate, the interaction is minimum whereas for 60° intersect significant interaction forces exist. The results in this manuscript are publishable in MDPI-Materials with minor suggested revision

  1. Dislocation-precipitate interaction is a classical topic that has been studied for many decades. The author could cite some recent reviews to highlight that this field of study continues to be an active topical area of research. See reference below, as an example (there may be others):

Precipitation in nanostructured alloys: A brief review, MRS Bulletin, vol. 46, March 2020, pages 250-257.

  1. Precipitate cutting also depends on temperature and strain rate andtransition state theory can be used to compute occurrence probability of a trigger event. See reference below:

Atomistic mechanism and probability determination of the cutting of Guinier-Preston zones by edge dislocations in dilute Al-Cu alloys, PHYSICAL REVIEW MATERIALS, FEB  2020.

DOI: 10.1103/PhysRevMaterials.4.020601

Reviewer 3 Report

The article addresses an interaction between dislocations and small
disc-shaped strengthening precipitates in aluminum alloys.

The introductory part summarizes the state of the art. In the following
section, the author focuses on the micromechanics, particularly on the stress
field due to an ellipsoidal inclusion (Eshelby inclusion problem), which is
solvable analytically, and a case of an arbitrary shape, which is solvable
numerically.

Furthermore, necessary parts of the dislocation theory are presented (stress
field due to a dislocation line, Peach-Koehler force). The thermodynamics of
precipitation hardening is discussed in detail.

The next section discusses the discrete dislocation dynamics (DDD). The 3D DDD
technique is discussed in author's earlier publications. Here is described the
interaction of a dislocation line (coming from a Frank-Read source) with a
disc-like precipitate. The author considers several orientations of the
precipitate. Depending on the orientation, the dislocation motion can be
retarded or promoted (60° angle between the Burgers vector and the
precipitate), or the interaction may be negligible (Burgers vector parallel to
the precipitate). The numerical solutions of the inclusion problem are
comparable with analytical solutions for the elliptical inclusion and improve
the accuracy.

In conclusion, the strengthening effect of the disc-shaped precipitates is
strongly orientation dependent.

I have enjoyed reading the manuscript and I would like to recommend it for
publication. The manuscript is written in good English, however, I would
suggest some spell checking (e.g. "such as" instead of "such by" on line 85,
"arbitrary" instead of "arbitral" on line 92).

I have also one more suggestion regarding the Figure 2. The arrows denoting
the interaction forces are difficult to read on a greyscale printout. A change
of the colour to e.g. red may help.

Reviewer 4 Report

This paper tackles an important issue which is the elastic interactions between dislocations and precipitates in aluminum alloys. Stress fields of misfit precipitates are computed from Green’s function method and then the motion of an initial straight screw dislocation with fixed ending points is simulated taking the external, dislocation segments and precipitate stress fields into account. {001} and {111} plates are considered. The methodology and the results of the simulations are quite interesting. However, I have the following remarks.

The English should be carefully checked and improved, for instance:

- C_ijkl is stiffness à C_ijkl is the stiffness tensor

- strait à straight

- week à weak

- burgers à Burgers

- clime forces à climb forces

- “interaction force of glide dislocation is made comparison to that with several different intersecting manners of dislocation and precipitate” (line 128): I cannot understand this sentence, in particular the expression “is made comparison to”

Symbols such as “GP-zones” should defined when using them for the first time.

Assumptions made about elasticity in the simulations should be stated clearly (linear, homogeneous/heterogeneous, isotropic/anisotropic) and the values of the elastic constants used should be provided.

The sentence “Furthermore, analytical solution for the stress outside Ω does not exist,..” in line 94 should be qualified. For instance, explicit analytical solutions of the exterior Eshelby tensor do exist under certain assumptions, for instance for an isotropic spheroidal inclusion (cf Ju and Sun, International Journal of Solids and Structures 38 (2001) 183-201).

In line 133, what is the “total plastic displacement” the author speaks about? Is it plastic strain?

For more clarity, I guess that \sigma^0_ij should be instead of \sigma^T_ij in the right-hand-side of Eq. 6. Is the external stress \sigma^0_ij assumed homogeneous? or can it vary with the position on the boundary?

In Eq. 8, I guess that f^T is the Peach-Koehler force related to the total stress. However, for consistency, this notation should have already been introduced in Eq. 5.

Eqs. 11 and 13 are hard to read due to superimposed prints.

The description of the “annihilation behavior” in pages 6 and 7 is not clear. I understand that the direction is “negative” but what quantities are really annihilated?

The author should justify the choice of \epsilon^*_33=0.04 for the eigenstrain and eventually add references. Besides, he mentioned two times that various misfit strains are considered (lines 9 and 64) but this does seem the case.

The value of the distance between the two ending points of the initial dislocation should be given.

Round 2

Reviewer 1 Report

The author has not made any material changes to the content of the manuscript, so my initial assessment holds.

Reviewer 4 Report

The author has adequately addressed the different issues raised by the reviewers.

Author Response

The author would like to mention great thanks for reviewer4, kind review of our manuscript is greatly acknowledged.